# The link between the atherogenic index of plasma and the risk of hypertension: Analysis from NHANES 2017–2020

Kaiyou Liu[1]*, Qingwei Ji[1], Shaoming Qin[1], Ling Liu[1], Hongwei Zhang[2], Huating Huang[1], Guihua Li[1], Junjun Chen[1]

1 People's Hospital of Guangxi Zhuang Autonomous Region, Nanning, China, 2 Youjiang Medical University for Nationalities, Baise, China

* lkyqrm@163.com

**Data Availability Statement:** All relevant data are within the manuscript and its Supporting Information files.

## Abstract

### Background

The atherogenic index of plasma (AIP) is a newly identified metabolic marker for atherosclerosis. However, there are inconsistent conclusions regarding the relationship between AIP and hypertension.

### Methods

The study subjects were sourced from the National Health and Nutrition Examination Survey (NHANES) database from 2017 to 2020. Logistic regression analyses were employed to explore the correlation between AIP and hypertension. The value of AIP in predicting hypertension was assessed using ROC curves, and their nonlinear relationship was described using restricted cubic splines (RCS). Subgroup analyses, interactions, and sensitivity analyses were also conducted.

### Results

The study included 7,067 participants who were sourced from the NHANES database. There were 2723 participants diagnosed hypertension. We observed a notable correlation between AIP and hypertension (OR:1.89, 95%CI: 1.11–3.22, P = 0.019). ROC curve showed AIP had a good predictive value for the onset of hypertension, with the AUC of 0.652 (95% CI:0.639–0.664, p<0.001). RCS found that there existed a nonlinear association between AIP and the incidence of hypertension (p<0.001). Even after excluding individuals under the age of 40 years old, the results still indicate a strong association between AIP and hypertension.

### Conclusions

AIP may serve as an early biological marker for identifying hypertension, facilitating early screening of susceptible populations.

**Funding:** The author(s) received no specific funding for this work.

**Competing interests:** The authors have declared that no competing interests exist.

## Introduction

With the advent of population aging, the prevalence of hypertension has been increasing. According to the 2023 WHO hypertension report, in the last three decades, the prevalence of hypertension among adults aged 30–79 has doubled, and is expected to reach 1.3 billion [1]. At the same time, only 54% of hypertension cases are diagnosed, 42% have received treatment, and only 21% have their blood pressure controlled [1]. As blood pressure rises, the incidence of coronary artery disease (CAD), heart failure (HF), and stroke will increase. Hypertension is an important and adjustable risk factor for cardiovascular and cerebrovascular diseases. Actively and effectively controlling blood pressure can reduce the risk of coronary heart disease, myocardial infarction, and stroke [2], and can also lower overall mortality and cardiovascular mortality rates [3,4]. Therefore, timely detection and treatment of hypertension is of significant public health importance.

The atherogenic index of plasma (AIP), integrates protective and atherogenic lipoproteins, serving as an indicator of lipid metabolism. AIP was initially proposed in 2000 [5], and has been closely associated with coronary heart disease [6,7], HF [8], obesity [9], diabetes mellitus (DM) [10]. However, there are relatively few articles exploring the link between AIP and hypertension, and the conclusions of related studies are inconsistent [11]. Research has shown a close relationship between AIP and hypertension as well as vascular sclerosis [12]. However, some studies suggest that AIP is not associated with blood pressure levels or systemic vascular resistance [13,14]. Therefore, we try to investigate the link between AIP and the incidence of hypertension by a large-scale database.

## Methods

### Study population

The data for this study were extracted from the National Health and Nutrition Examination Survey (NHANES) database from 2017 to 2020. NHANES is structured to evaluate the health and nutritional status of the American populace. Detailed information on NHANES study design and data can be found at http://www.cdc.gov/nchs/nhanes.htm. The study initially enrolled 9,445 individuals. We excluded participants who were (1) under 18 years old, (2) missing serum TG or HDL-C values, and (3) absence of hypertension status. Individuals lacking information on study variables such as age, smoking, BMI, and conditions like CAD, DM, and HF were also excluded. Ultimately, 7,067 individuals met the criteria (Fig 1). Due to NHANES being a publicly available database and the related data lacking personal identifying information, this study does not require approval from an ethics committee.

### Assessment of hypertension

The occurrence of hypertension was determined by self-reporting "Yes" in response to the MCQ questionnaire, which asked, "Has a doctor or other healthcare provider ever informed you that you have hypertension?".

### Definition of atherogenic index of plasma

The formula for AIP involves taking the base 10 logarithm of (TG/HDL-C), reflecting the logarithmic association between TG and HDL-C [5].

### Covariates definition

We extracted variables from the NHANES database, including age, sex, race, education level, BMI, diabetes mellitus (DM), HF and CAD. The diagnosis of CAD, DM, and HF was based on

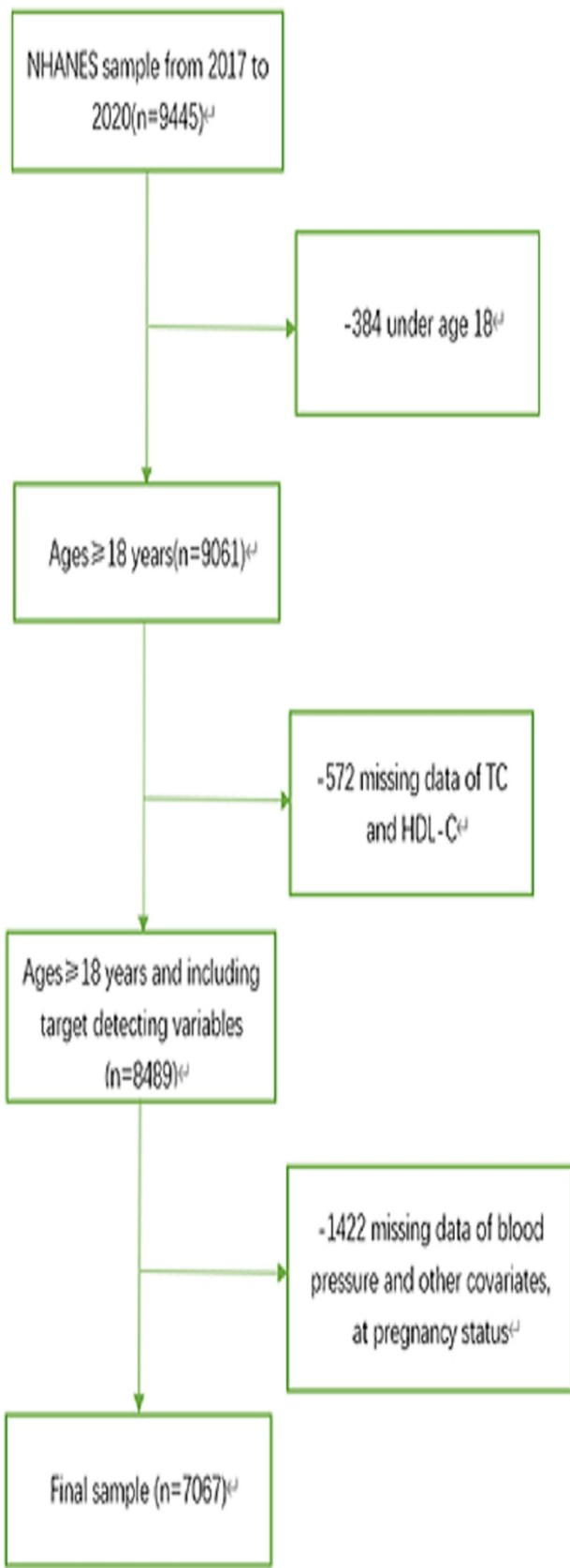

**Fig 1. Flowchart of the sample selection from NHANES 2017–2020.**

questionnaire responses to "Ever told you had XX?" and recorded as "Yes/No". Smoking status was dichotomized as "Yes/No" based on the question "Smoked at least 100 cigarettes in life?". Alcohol consumption was dichotomized as "Yes/No" based on the question "Ever had 4/5 or more drinks every day?". Additionally, other laboratory testing indicators were collected, including blood glucose, blood lipids, liver function, kidney function, blood uric acid, and other indicators.

## Statistical analyses

The R software (R version 4.3.1) and SPSS (22.0) were utilized for our statistical analysis.

The constant variables were documented with the mean±standard deviation, and use t-tests or analysis of variance (ANOVA) for statistics. Categorical variables are encoded numerically (as proportions) and subjected to analysis using chi-square tests. We employed logistic regression analyses to examine the link between AIP and hypertension, calculating odds ratios (OR) with 95% confidence intervals (95% CI) to identify independent risk factors for hypertension. We assessed the predictive value of AIP for hypertension by the area under the receiver operating curve (ROC). An RCS assay was employed for investigating whether AIP is correlated with hypertension occurrence dose-responsively. Finally, subgroup analysis was used, patients were separated into subgroups based on their gender, drinking, smoking, race, CAD status, DM status and HF status. P-values < 0.05 was regarded as indicating statistical significance.

## Results

### Characteristics of participants

In this study, a total of 7,067 patients were finally included, with males comprising 49.35%. Among them, there were 2,723 cases of hypertension patients, with an average age of 59.98 ± 14.15 years. Non-hypertensive patients numbered 4,344, with an average age of 45.11 ± 16.64 years. Compared to non-hypertensive patients, those with hypertension were older, had a higher smoking rate, and exhibited higher BMI, HbA1c, TG, UA, and Scr levels. Moreover, they had a higher probability of concomitant CAD, HF, and DM (P<0.001). Additionally, the AIP index between the two groups showed statistically significant differences (Table 1).

### Logistic regression analysis of AIP and hypertension risk

Regression analysis showed that age, BMI, HbA1c, TG, FBG, Scr, and AIP were significantly associated with hypertension. In Model 2, adjusting for gender and race, and in Model 3, adjusting for gender, race, alcohol consumption, DM, smoking, HF, and CAD, the AIP index remained significantly associated with hypertension (OR: 1.89, 95% CI: 1.11–3.22, P = 0.019 in Model 3) (Table 2).

### RCS plots and ROC analyses of AIP and hypertension risk

According to our RCS results, AIP was non-linearly correlated with the incidence of hypertension (p<0.001) (Fig 2). As shown in Fig 3, AIP possessed an acceptable AUC value of 0.652 (95% CI:0.639–0.664, p<0.001) in calculating hypertension onset in our ROC analysis, with a corresponding specificity and sensitivity of 58.3% and 53.3%, respectively.

### Subgroup analysis

Subgroup analyses and interaction tests were conducted among patients with hypertension after categorizing them according to their gender, drinking, smoking, race, CAD status, DM

**Table 1. Baseline characteristics of participants.**

| Variables | No hypertension(n = 4344) | Hypertension(n = 2723) | P value |
|---|---|---|---|
| Gendle(male) | 2117(48.73) | 1371 (50.35) | 0.186 |
| Age (years) | 45.11 ± 16.64 | 59.98 ± 14.15 | < .001 |
| Smoking [(n%)] | 1661 (38.24) | 1317 (48.37) | < .001 |
| Drinking [(n%)] | 3980 (91.62) | 2504 (91.96) | 0.617 |
| HF [(n%)] | 46 (1.06) | 211 (7.75) | < .001 |
| CAD [(n%)] | 65 (1.50) | 247 (9.07) | < .001 |
| DM [(n%)] | 418 (9.62) | 868 (31.88) | < .001 |
| Race [(n%)] | | | < .001 |
| Mexican American | 602 (13.86) | 202 (7.42) | |
| Non-Hispanic Asian | 470 (10.82) | 248 (9.11) | |
| Non-Hispanic Black | 1537 (35.38) | 1018 (37.39) | |
| Non-Hispanic White | 939 (21.62) | 888 (32.61) | |
| Other Hispanic | 580 (13.35) | 239 (8.78) | |
| Other Race | 216 (4.97) | 128 (4.70) | |
| SBP(mmHg) | 119.26 ± 16.51 | 133.57 ± 20.85 | < .001 |
| DBP(mmHg) | 73.31 ± 10.34 | 77.97 ± 13.22 | < .001 |
| HBA1c(%) | 5.66 ± 0.99 | 6.17 ± 1.25 | < .001 |
| LDL-C(mmol/L) | 2.84 ± 0.64 | 2.77 ± 0.67 | < .001 |
| TC(mmol/L) | 4.85 ± 1.04 | 4.74 ± 1.09 | < .001 |
| TG(mmol/L) | 1.49 ± 1.23 | 1.69 ± 1.16 | < .001 |
| HDL-C(mmol/L) | 1.40 ± 0.41 | 1.36 ± 0.41 | < .001 |
| BMI (kg/m$^2$) | 28.94 ± 6.98 | 31.87 ± 7.58 | < .001 |
| ALT(U/L) | 22.21 ± 17.82 | 22.41 ± 20.23 | 0.673 |
| AST(U/L) | 21.73 ± 13.18 | 22.20 ± 15.65 | 0.180 |
| Scr (mmol/L) | 75.06 ± 23.93 | 88.54 ± 64.88 | < .001 |
| AIP | -0.03 ± 0.32 | 0.05 ± 0.31 | < .001 |
| UA(umol/L) | 308.36 ± 82.51 | 344.03 ± 91.95 | < .001 |

HF: Heart failure; CAD: Coronary artery disease; DM: Diabetes mellitus; SBP: Systolic blood pressure; DBP: Diastolic blood pressure; LDL-C: Low-density lipoprotein cholesterol; TC: Total cholesterol; TG: Triglycerides; HDL-C: High-density lipoprotein cholesterol; BMI: Body mass index; ALT: Alanine aminotransferase; AST: Aspartate aminotransferase; Scr: Serum creatinine; AIP: Atherogenic index of plasma; UA: Uric acid.

status and HF status. We observed that there was no interaction between AIP and the above-mentioned variables (Fig 4).

## Sensitive analysis

In order to enhance the reliability of our study findings, after excluding participants under 40 years old, we conducted the analysis again. After adjusting for relevant variables, we continued to observe a notable correlation between AIP and hypertension(OR:2.13,95%CI: 1.17–3.90, P = 0.014)(S1 Table).

## Discussion

Hypertension has a high prevalence, with approximately 1.3 billion patients worldwide. It is well-known that hypertension risk factors are diverse, including age, gender, obesity, high-salt and high-fat diets, and family history. Therefore, early and accurate identification of high-risk population for hypertension is crucial for timely intervention.

**Table 2. Univariate and multivariate analyses between AIP and hypertension.**

|  | Model 1 | | Model 2 | | Model 3 | |
|---|---|---|---|---|---|---|
|  | OR (95%CI) | P | OR (95%CI) | P | OR (95%CI) | P |
| HDL-C | 0.78(0.69–0.88) | < .001 | 1.66(1.25–2.20) | < .001 | 1.54(1.16–2.05) | 0.003 |
| Age | 1.06(1.06–1.06) | < .001 | 1.05(1.05–1.06) | < .001 | 1.05(1.04–1.05) | < .001 |
| HBA1c | 1.57(1.49–1.66) | < .001 | 1.14(1.05–1.24) | 0.003 | 0.98(0.89–1.07) | 0.669 |
| SBP | 1.04(1.04–1.05) | < .001 | 1.02(1.01–1.03) | < .001 | 1.02(1.01–1.03) | < .001 |
| DBP | 1.04(1.03–1.04) | < .001 | 1.02(1.02–1.03) | < .001 | 1.02(1.02–1.03) | < .001 |
| LDL-C | 0.84(0.78–0.91) | < .001 | 1.05(0.93–1.19) | 0.396 | 1.08(0.95–1.22) | 0.239 |
| TC | 0.91(0.87–0.95) | < .001 | 0.75(0.68–0.82) | < .001 | 0.79(0.72–0.87) | < .001 |
| BMI | 1.06(1.05–1.06) | < .001 | 1.05(1.04–1.06) | < .001 | 1.05(1.04–1.06) | < .001 |
| ALT | 1.00(1.00–1.00) | 0.673 | 1.00(1.00–1.01) | 0.339 | 1.00(1.00–1.01) | 0.160 |
| AST | 1.00(1.00–1.01) | 0.184 | 1.00(0.99–1.01) | 0.595 | 1.00(0.99–1.00) | 0.369 |
| Scr | 1.02(1.01–1.02) | < .001 | 1.01(1.01–1.01) | < .001 | 1.01(1.01–1.01) | 0.001 |
| UA | 1.01(1.01–1.01) | < .001 | 1.02(1.01–1.01) | < .001 | 1.01(1.01–1.01) | < .001 |
| FBG | 1.21(1.18–1.25) | < .001 | 1.02(0.98–1.07) | 0.325 | 1.02(0.98–1.07) | 0.333 |
| TG | 1.15(1.10–1.20) | < .001 | 0.98(0.89–1.08) | 0.694 | 0.99(0.90–1.09) | 0.847 |
| AIP | 2.14(1.84–2.49) | < .001 | 2.20(1.30–3.74) | 0.003 | 1.89(1.11–3.22) | 0.019 |

SBP: Systolic blood pressure; DBP: Diastolic blood pressure; LDL-C: Low-density lipoprotein cholesterol; TC: Total cholesterol; TG: Triglycerides; HDL-C: High-density lipoprotein cholesterol; BMI: Body mass index; ALT: Alanine aminotransferase; AST: Aspartate aminotransferase; Scr: Serum creatinine; AIP: Atherogenic index of plasma; UA: Uric acid; FBG: Fasting blood sugar; OR: Odds ratio; CI: Confidence intervals.

In this study, we included 7,067 NHANES participants from 2017 to 2020. The results suggest that AIP, as an indicator reflecting lipid metabolism, has significant predictive value for the occurrence of hypertension and plays an important role in hypertension prevention.

This study marks the initial utilization of the NHANES database to explore the correlation between AIP and hypertension. Although our results indicate that the sensitivity and specificity of AIP in predicting hypertension are 53.3% and 58.3%, respectively, with an AUC of 65.2%, our study boasts a relatively large sample size. Additionally, we conducted subgroup analyses and sensitivity analyses to further validate the relationship between AIP and hypertension.

Our study is consistent with previous findings in other populations [15,16], In the cross-sectional study[15], Tan et al. evaluated 15,453 participants aged 18 or above with normal blood glucose levels from Gifu, Japan. The results revealed an increased risk of prehypertension and hypertension among participants with high levels of AIP, this association was particularly prominent among female patients aged 40 to 60. Yuan et al. [16] reported a nationwide cohort study conducted in China, which included a total of 3,150 participants. After six years of follow-up, it was found that 1,054 of these participants developed hypertension. It revealed a positive correlation between AIP and the incidence of hypertension. These two articles align with ours in possessing large sample sizes, which ensures the credibility of the research findings. However, another study[14] suggests that AIP is not associated with hypertension, the research relied on measuring office blood pressure to diagnose hypertension, and it involved only a total of 615 patients, among which 366 had elevated blood pressure, these factors may have led to inconsistencies in its results.

AIP, as a biomarker reflecting lipid metabolism, is significantly associated with cardiovascular diseases, metabolic syndrome, and related conditions. Previous studies have found that AIP is correlated with adverse prognosis in coronary heart disease patients [17], and it is highly correlated with the instability of coronary artery plaques [18]. Additionally, research has

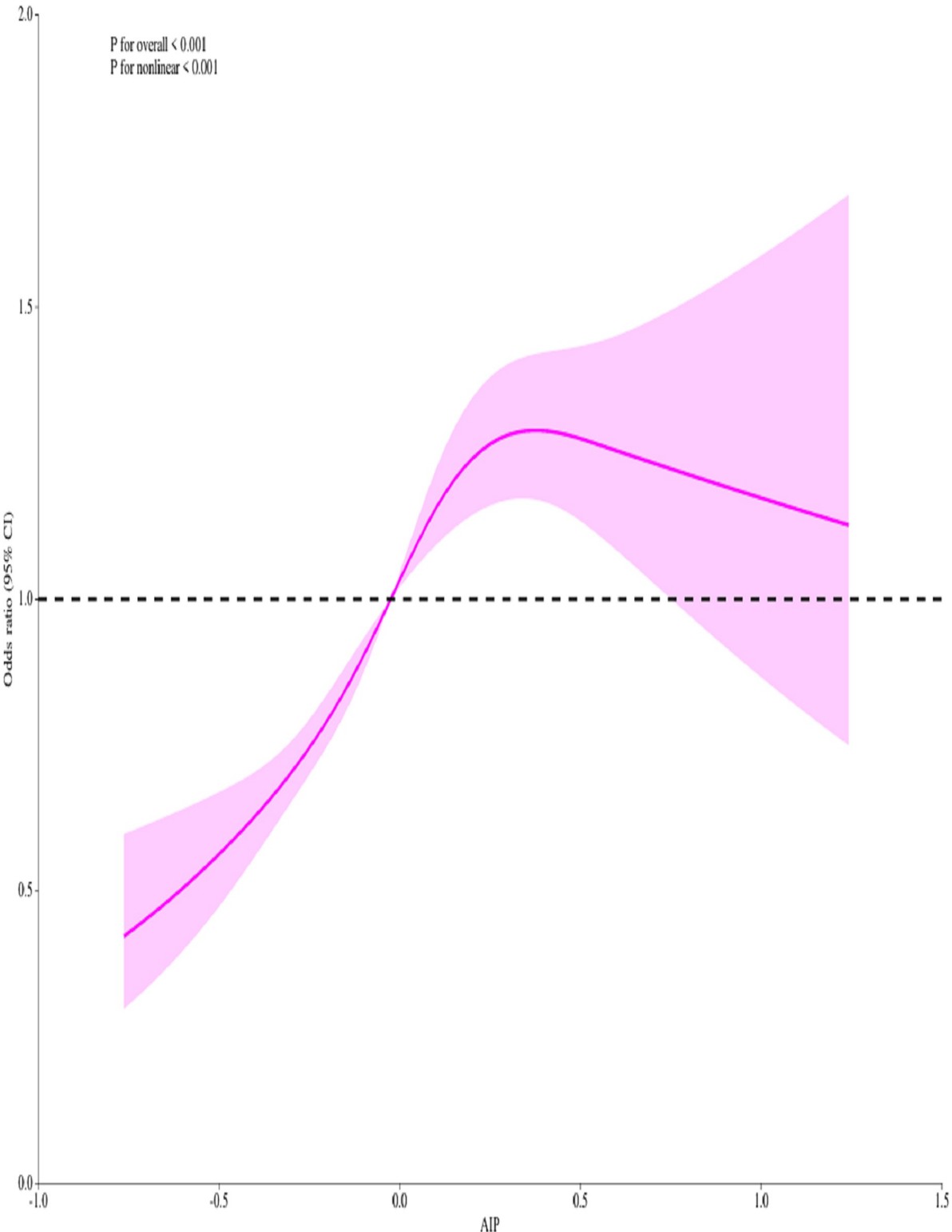

**Fig 2. The analysis of restricted cubic spline regression.**

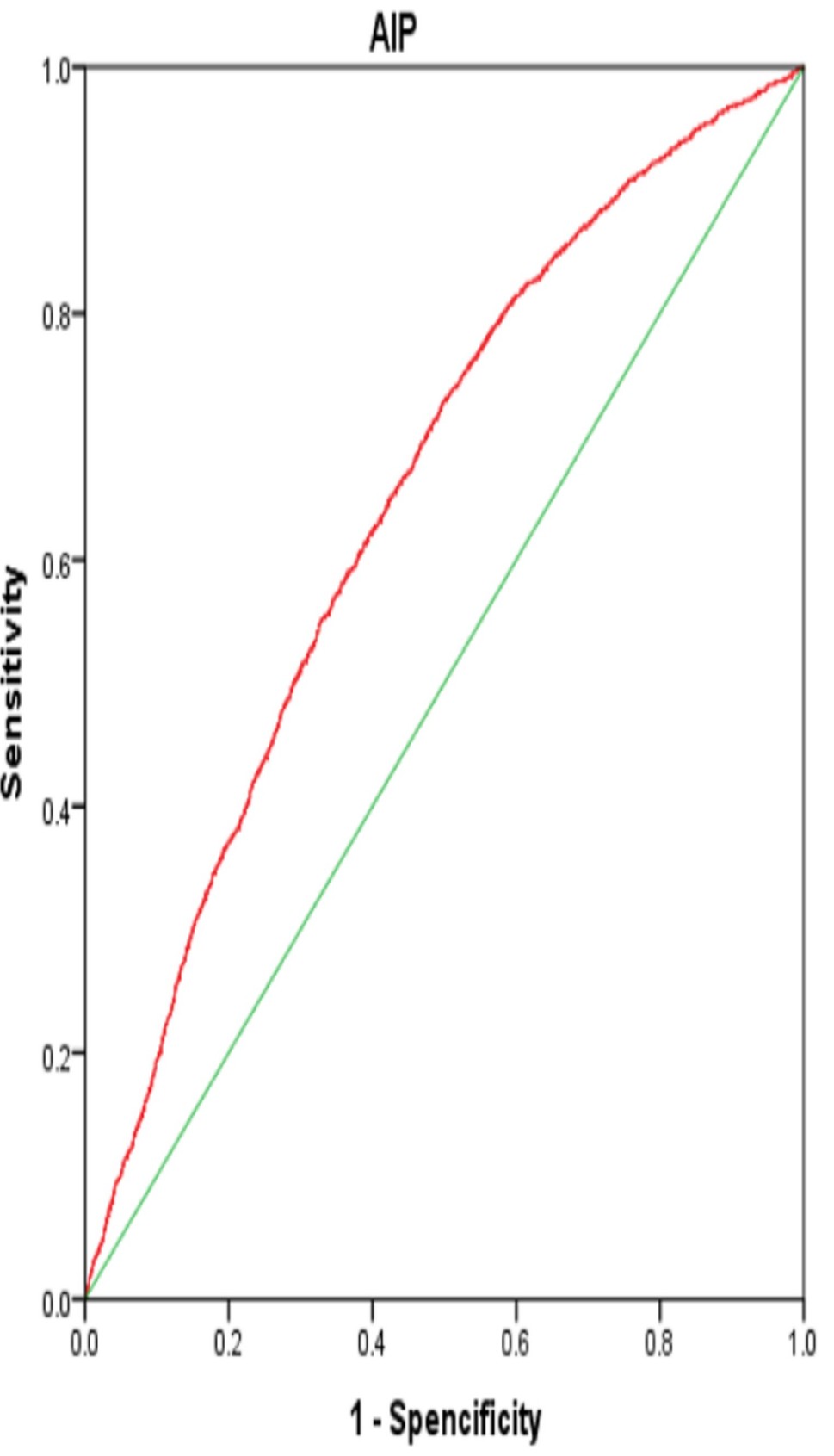

**Fig 3. The analysis of ROC curve.**

| Variables | n (%) | AIP < -0.025 | AIP ≥ -0.025 | OR (95% CI) | | P for interaction |
|---|---|---|---|---|---|---|
| All patients | 7067 (100.00) | 1136/3450 | 1587/3617 | 1.53 (1.38 ~ 1.70) | | |
| Gender | | | | | | 0.075 |
| Female | 3579 (50.64) | 638/2031 | 714/1548 | 1.73 (1.49 ~ 2.02) | | |
| Male | 3488 (49.36) | 498/1419 | 873/2069 | 1.42 (1.23 ~ 1.65) | | |
| Drink | | | | | | 0.771 |
| No | 583 (8.25) | 86/255 | 133/328 | 1.53 (1.05 ~ 2.24) | | |
| Yes | 6484 (91.75) | 1050/3195 | 1454/3289 | 1.54 (1.38 ~ 1.71) | | |
| DM | | | | | | 0.268 |
| No | 5781 (81.80) | 856/3032 | 999/2749 | 1.34 (1.19 ~ 1.51) | | |
| Yes | 1286 (18.20) | 280/418 | 588/868 | 1.11 (0.86 ~ 1.43) | | |
| Smoke | | | | | | 0.987 |
| No | 4089 (57.86) | 621/2128 | 785/1961 | 1.47 (1.28 ~ 1.70) | | |
| Yes | 2978 (42.14) | 515/1322 | 802/1656 | 1.50 (1.29 ~ 1.75) | | |
| HF | | | | | | 0.256 |
| No | 6810 (96.36) | 1063/3354 | 1449/3456 | 1.47 (1.32 ~ 1.64) | | |
| Yes | 257 (3.64) | 73/96 | 138/161 | 1.98 (1.03 ~ 3.80) | | |
| CAD | | | | | | 0.394 |
| No | 6755 (95.59) | 1055/3347 | 1421/3408 | 1.48 (1.33 ~ 1.65) | | |
| Yes | 312 (4.41) | 81/103 | 166/209 | 1.08 (0.60 ~ 1.95) | | |
| Race | | | | | | 0.124 |
| Mexican American | 804 (11.38) | 50/303 | 152/501 | 1.83 (1.23 ~ 2.71) | | |
| Non-Hispanic Asian | 819 (11.59) | 75/358 | 164/461 | 1.74 (1.23 ~ 2.46) | | |
| Non-Hispanic Black | 1827 (25.85) | 527/1183 | 361/644 | 1.51 (1.23 ~ 1.85) | | |
| Non-Hispanic White | 2555 (36.15) | 354/1153 | 664/1402 | 2.09 (1.75 ~ 2.48) | | |
| Other Hispanic | 718 (10.16) | 85/291 | 163/427 | 1.24 (0.87 ~ 1.76) | | |
| Other Race | 344 (4.87) | 45/162 | 83/182 | 2.21 (1.36 ~ 3.58) | | |

**Fig 4. The subgroup analysis.**

shown that individuals with high AIP levels have increased risks of obesity [9,19], diabetes, and cardiovascular metabolic disorders [20,21]. Recent studies have discovered a link between AIP and genetic polymorphism of CETP rs708272, males possessing the minor T allele exhibit lower AIP levels than those with the CC genotype [22].

The potential mechanisms linking AIP and hypertension are still under further investigation, but they are inclined towards factors such as lipid deposition, insulin resistance, adipokine secretion, and autonomic nervous system dysfunction [23,24]. Additionally, studies have found that HDL-C may contribute to the regulation of endogenous nitric oxide bioavailability, when HDL-C levels decrease, the bioavailability of endogenous nitric oxide decreases, leading to endothelial dysfunction and consequently hypertension [25,26]. Additionally, abnormal lipid levels promote atherosclerosis, reducing arterial vessel wall compliance and consequently causing hypertension [27,28]. Reactive oxygen species are pivotal in the pathogenesis of hypertension, studies indicate that lowering levels of reactive oxygen species can effectively control hypertension [29]. High-carbohydrate and high-fat diets increase blood lipid levels and elevate oxidative stress levels in the body, leading to excessive production of reactive oxygen species [30], which in turn triggers long-term oxidative stress and inflammatory responses in cells and organs, thereby causing diseases such as hypertension [30].

This article has some shortcomings. Firstly, it is a cross-sectional design, which can only collect information on the relationship between relevant biomarkers and hypertension but cannot establish causal relationships. Secondly, it only collected baseline clinical data like AIP, without assessing whether fluctuations in AIP are associated with blood pressure. Additionally, we relied on self-reported participant data to identify occurrences of conditions such as hypertension, CAD, and HF, which may lead to some cases being missed or misdiagnosed.

## Conclusion

In summary, this study utilized large-scale data from public databases to find that individuals with higher AIP indices have a higher risk of developing hypertension. Therefore, we can use AIP for early screening of high-risk populations, thereby improving early prevention and related interventions.

## Supporting information

**S1 Table. Logistic analyses between AIP and hypertension after excluding participants younger than 40.**
(DOCX)

**S1 File.**
(XLSX)

## Acknowledgments

We express our gratitude to the team which is responsible for collecting and maintaining the NHANES database. It is through their efforts that we are able to utilize this database.

## Author Contributions

**Data curation:** Kaiyou Liu, Guihua Li, Junjun Chen.

**Formal analysis:** Kaiyou Liu, Guihua Li, Junjun Chen.

**Methodology:** Huating Huang.

**Software:** Hongwei Zhang.

**Supervision:** Qingwei Ji.

**Visualization:** Hongwei Zhang.

**Writing – original draft:** Kaiyou Liu.

**Writing – review & editing:** Qingwei Ji, Shaoming Qin, Ling Liu.

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
