## [Decision Letter · Decision Letter 0]

8 Nov 2024

PONE-D-24-30105The link between the atherogenic index of plasma and the risk of hypertension:Analysis from NHANES 2017–2020PLOS ONE

Dear Dr. Liu,

Thank you for submitting your manuscript to PLOS ONE. After careful consideration, we feel that it has merit but does not fully meet PLOS ONE’s publication criteria as it currently stands. Therefore, we invite you to submit a revised version of the manuscript that addresses the points raised during the review process.This manuscript requires a minor revisionCarefully and thoroughly address all the comments from the reviewersThe details of reviewers concerns are found below. Please read them carefully, address them and revise the manuscript accordingly. 

We look forward to receiving your revised manuscript.

Kind regards,

Fredirick Lazaro mashili, MD, PhD

Academic Editor

PLOS ONE

Additional Editor Comments:

Please respond thoroughly all the comments given by the reviewers

Reviewers' comments:

Reviewer's Responses to Questions

**Comments to the Author**

1. Is the manuscript technically sound, and do the data support the conclusions?

Reviewer #1: Yes

Reviewer #2: Yes

2. Has the statistical analysis been performed appropriately and rigorously? 

Reviewer #1: Yes

Reviewer #2: Yes

3. Have the authors made all data underlying the findings in their manuscript fully available?

Reviewer #1: Yes

Reviewer #2: Yes

4. Is the manuscript presented in an intelligible fashion and written in standard English?

Reviewer #1: Yes

Reviewer #2: Yes

5. Review Comments to the Author

Reviewer #1: This manuscript discusses results from comprehensive analysis of robust data. Generally, the methods (including statistical approach) are sound, results well described and the manuscript is very well written. The disussion though well written, is not comprehensive enough. The authors main reason to undertake the analysis was to help in ressolving previous conflicting results regarding the use of atherogenic index of plasma in predicting hypertension. While the authors have done a good job in writing the discussion, they did not discuss possible reasons for conflicting results. The authors should highlight and discuss any methodological differences between studies with conflicting results and theirs, to bring to light any possible reason for the conflicting results.

Reviewer #2: The topic is highly novel, and the methodology is accurately implemented. However, the author needs to clarify and elaborate on the rationale behind stating that AIP is a good predictor of hypertension, given the sensitivity and specificity values of 53.3% and 58.3%, respectively. Additionally, the AUC of 65.2% does not strongly support AIP as an effective predictor of hypertension.

In the discussion section, the author should begin by summarising the key findings of the study. Following this, the results should be discussed compared and contrasted with findings from other studies, explaining the potential reasons for similarities or differences. More literature on AIP and hypertension should be included on introduction section, while avoiding an extensive literature review in the discussion part.

6. PLOS authors have the option to publish the peer review history of their article (what does this mean?). If published, this will include your full peer review and any attached files.

Reviewer #1: **Yes: **Fredirick Mashili

Reviewer #2: **Yes: **Ikunda Dionis

---

## [Author Response · Author response to Decision Letter 0]

1 Dec 2024

Dear Editors and Reviewers: 

 Thank you for your letter and for the reviewers’ comments concerning our manuscript entitled " The link between the atherogenic index of plasma and the risk of hypertension：Analysis from NHANES 2017–2020". Those comments are all valuable and very helpful for revising and improving our paper. We have studied comments carefully and have made correction which we hope meet with approval. Revised portion are marked in red in the paper. The main corrections in the paper and the responds to the reviewer’s comments are as follows: 

Responds to the reviewer’s comments: 

Reviewer #1: The discussion though well written, is not comprehensive enough. The authors main reason to undertake the analysis was to help in resolving previous conflicting results regarding the use of atherogenic index of plasma in predicting hypertension. While the authors have done a good job in writing the discussion, they did not discuss possible reasons for conflicting results. The authors should highlight and discuss any methodological differences between studies with conflicting results and theirs, to bring to light any possible reason for the conflicting results.

Response: We are very sorry for our negligence of the discussion of conflicting results, we have made correction according to the Reviewer’s comments. We have discussed the characteristics of these studies and the potential reasons for the inconsistent results in lines 173 to 185.

Reviewer #2: The topic is highly novel, and the methodology is accurately implemented. However, the author needs to clarify and elaborate on the rationale behind stating that AIP is a good predictor of hypertension, given the sensitivity and specificity values of 53.3% and 58.3%, respectively. Additionally, the AUC of 65.2% does not strongly support AIP as an effective predictor of hypertension.

Response: As Reviewer suggested that the sensitivity and specificity values of 53.3% and 58.3%, respectively. Additionally, the AUC of 65.2% does not strongly support AIP as an effective predictor of hypertension. But our study boasts a relatively large sample size. Additionally, we conducted subgroup analyses and sensitivity analyses to further validate the relationship between AIP and hypertension.

We tried our best to improve the manuscript and made some changes in the manuscript. And here we did not list the changes but marked in red in revised paper. We appreciate for Editors/Reviewers’ warm work earnestly, and hope that the correction will meet with approval. Once again, thank you very much for your comments and suggestions.

Best regards!

Yours sincerely, Kaiyou Liu

Corresponding author: Kaiyou Liu

E-mail: lkyqrm@163.com

---

## [Decision Letter · Decision Letter 1]

22 Dec 2024

The link between the atherogenic index of plasma and the risk of hypertension:Analysis from NHANES 2017–2020

PONE-D-24-30105R1

Dear Dr. Liu,

We’re pleased to inform you that your manuscript has been judged scientifically suitable for publication and will be formally accepted for publication once it meets all outstanding technical requirements.

Kind regards,

Fredirick Lazaro mashili, MD, PhD

Academic Editor

PLOS ONE

Additional Editor Comments (optional):

All the comments have been sufficiently addressed.

Reviewers' comments:

Reviewer's Responses to Questions

**Comments to the Author**

1. If the authors have adequately addressed your comments raised in a previous round of review and you feel that this manuscript is now acceptable for publication, you may indicate that here to bypass the “Comments to the Author” section, enter your conflict of interest statement in the “Confidential to Editor” section, and submit your "Accept" recommendation.

Reviewer #1: All comments have been addressed

Reviewer #2: All comments have been addressed

2. Is the manuscript technically sound, and do the data support the conclusions?

Reviewer #1: Yes

Reviewer #2: Yes

3. Has the statistical analysis been performed appropriately and rigorously? 

Reviewer #1: Yes

Reviewer #2: Yes

4. Have the authors made all data underlying the findings in their manuscript fully available?

Reviewer #1: Yes

Reviewer #2: Yes

5. Is the manuscript presented in an intelligible fashion and written in standard English?

Reviewer #1: Yes

Reviewer #2: Yes

6. Review Comments to the Author

Reviewer #1: The authors have sufficiently addressed all the previously raised concerns. The manuscript fully adhere to journal's formatting requirements

Reviewer #2: All comments have been addressed. The manuscript is well-written and clearly articulated, with a robust sample size and thoroughly documented results.

7. PLOS authors have the option to publish the peer review history of their article (what does this mean?). If published, this will include your full peer review and any attached files.

Reviewer #1: **Yes: **Fredirick mashili

Reviewer #2: No

---

## [Editor Report · Acceptance letter]

16 Jan 2025

PONE-D-24-30105R1 

PLOS ONE

Dear Dr. Liu, 

I'm pleased to inform you that your manuscript has been deemed suitable for publication in PLOS ONE. Congratulations! Your manuscript is now being handed over to our production team.

Kind regards, 

on behalf of

Dr Fredirick Lazaro mashili 

Academic Editor

PLOS ONE